# Rectal and Tracheal Carriage of Carbapenemase Genes and Class 1 and 2 Integrons in Patients in Neurosurgery Intensive Care Unit

**DOI:** 10.3390/antibiotics11070886

**Published:** 2022-07-03

**Authors:** Ekaterina S. Kuzina, Tatiana S. Novikova, Evgeny I. Astashkin, Galina N. Fedyukina, Angelina A. Kislichkina, Natalia V. Kurdyumova, Ivan A. Savin, Olga N. Ershova, Nadezhda K. Fursova

**Affiliations:** 1Department of Training and Improvement of Specialists, State Research Center for Applied Microbiology and Biotechnology, Territory “Kvartal A”, 142279 Obolensk, Russia; e.leonova@mail.ru; 2Department of Molecular Microbiology, State Research Center for Applied Microbiology and Biotechnology, Territory “Kvartal A”, 142279 Obolensk, Russia; pozitifka.15@yandex.ru (T.S.N.); info@obolensk.org (E.I.A.); 3Department of Immunobiochemistry of Pathogenic Microorganisms, State Research Center for Applied Microbiology and Biotechnology, Territory “Kvartal A”, 142279 Obolensk, Russia; galafed@mail.ru; 4Department of Culture Collection, State Research Center for Applied Microbiology and Biotechnology, Territory “Kvartal A”, 142279 Obolensk, Russia; angelinakislichkina@yandex.ru; 5Department of Clinical Epidemiology, National Medical Research Center of Neurosurgery Named after Academician N.N. Burdenko, 125047 Moscow, Russia; NKurdumova@nsi.ru (N.V.K.); info@nsi.ru (I.A.S.); oershova@nsi.ru (O.N.E.)

**Keywords:** rectal and tracheal carriage, carbapenemase genes, class 1 and 2 integrons, neurosurgery, intensive care unit, Gram-negative bacteria, *Klebsiella pneumoniae*, *Escherichia coli*, *Acinetobacter baumannii*, *Pseudomonas aeruginosa*

## Abstract

The spread of multidrug-resistant Gram-negative bacteria, which is associated with the distribution of beta-lactamase genes and class 1 and 2 integrons, is a global problem. In this study, in the Moscow neurosurgery intensive care unit (neuro-ICU), the high prevalence of the above-stated genes was found to be associated with intestinal and tracheal carriage. Seven-point prevalence surveys, which included 60 patients in the neuro-ICU, were conducted weekly in the period from Oct. to Nov. 2019. A total of 293 clinical samples were analyzed, including 146 rectal and 147 tracheal swabs; 344 Gram-negative bacteria isolates were collected. Beta-lactamase genes (n = 837) were detected in the isolates, including beta-lactamase *bla*_TEM_ (n = 162), *bla*_SHV_ (n = 145), cephalosporinase *bla*_CTX–M_ (n = 228), carbapenemase *bla*_NDM_ (n = 44), *bla*_KPC_ (n = 25), *bla*_OXA–48_ (n = 126), *bla*_OXA–51–like_ (n = 54), *bla*_OXA–40-like_ (n = 43), *bla*_OXA–23-like_ (n = 8), and *bla*_VIM_ (n = 2), as well as class 1 (n = 189) and class 2 (n = 12) integrons. One extensively drug-resistant *Klebsiella pneumoniae* strain (sequence type ST39 and capsular type K23), simultaneously carried beta-lactamase genes, *bla*_SHV–40_ and *bla*_TEM–1B_, three carbapenemase genes, *bla*_NDM_, *bla*_KPC_, and *bla*_OXA–48_, the cephalosporinase gene *bla*_CTX–M_, and two class 1 integrons. Before this study, such heavily armed strains have not been reported, suggesting the ongoing evolution of antibiotic resistance.

## 1. Introduction

Antimicrobial resistance is one of the most serious threats to global health care [1,2]. In the last two decades, an increased number of infections caused by multidrug-resistant Gram-negative bacteria (MDR-GNB) have been reported [3,4]. As a result, increases in morbidity and mortality rates have been observed, as well as a rise in health care costs [5]. Carbapenems were the most effective antibiotic therapy for infections caused by MDR-GNB until the increased prevalence of carbapenem resistance was described in clinical settings [6]. The widespread dissemination of carbapenem-resistant Gram-negative bacteria (CR-GNB) is associated with carbapenemase genes in several functional classes (class A (*bla*_KPC_), class B (*bla*_NDM_ and *bla*_VIM_), and class D (*bla*_OXA-48_)) and the decreasing efficacy of carbapenem. In the case of CR-GNB, only some “second-line” drugs, such as polymyxins, tigecycline, and fosfomycin, and some “last resort” antibiotics, such as aztreonam/avibactam, ceftazidime/avibactam, meropenem/vaborbactam, and imipenem/relebactam, have demonstrated their effectiveness in some cases [7,8,9]. CR-GNB, including *Klebsiella pneumoniae*, *Escherichia coli*, *Acinetobacter baumannii,* and *Pseudomonas aeruginosa,* pose a significant public health threat [10]. These bacteria are members of the ESKAPE group of pathogens (*Enterobacter* spp., *Staphylococcus aureus*, *Klebsiella* spp., *Acinetobacter baumannii*, *Pseudomonas aeruginosa*, and *Enterococcus* spp.); this group notoriously causes the most health-care-associated infections and carries patterns of antimicrobial resistance [11,12]. The risk of CR-GNB acquisition has increased four-fold with carbapenem exposure, and new meta-analyses have confirmed excess mortality associated with such bacteria [3]. The trend toward an increase in the resistance of Gram-negative species among the ESKAPE group is of great concern, while it has been shown that the resistance of Gram-positive bacteria remains almost the same [13].

Recently, the asymptomatic carriage of CR-GNB and carbapenem resistance genes (CRGs) has been described in hospitalized patients as well as in healthy people in many countries [14,15,16,17,18]. The colonization of CR-GNB and CRGs has led to the development of nosocomial infections, an increase in the bloodstream infections, and a four-fold increase in mortality rate [3,19]. It has been suggested that a solution to this problem could be based on compliance with infection control measures such as reducing contacts between patients in the ICU and the utilization of the one-patient–one-nurse approach, rapid diagnostic techniques, and optimization treatment schemes [20,21,22,23]. To our knowledge, in Russia, little is known about the MDR-GNB and CRG colonization rates in hospitalized patients or in the wider community. Over the past 15 years, several single and multicenter microbiological surveillance studies regarding MDR-GNB hospital-acquired infections have been conducted in the Russian Federation [24,25,26,27,28,29]. These studies only contained descriptive reports and did not highlight the mechanisms of antimicrobial resistance or the colonization of patients via MDR-GNB and CRGs. Recently, the fecal carriage of *Klebsiella pneumoniae* ST23 was described in healthy persons in Russia [30].

Previous studies have shown that surgical antibiotic prophylaxis, commonly used in surgical hospitals, can influence the landscape of clinical isolates and their resistomes; therefore, prophylaxis protocols must be regularly reviewed on the basis of epidemiological surveys estimating patients’ colonization, especially by multidrug-resistant bacteria [31]. Screening for asymptomatic CR-GNB colonization and implementing contact precautions will reduce patient-to-patient transmission [23].

In order to provide possible future interventions for clinically rational antibiotic usage, in this study, our aim was to evaluate the rate of the rectal and tracheal carriage of MDR-GNB and CRGs in patients in the Moscow neuro-ICU. The novelty in our research is the description of “heavily armed” CR-GNB strains that were not reported prior to this study. The results of this study should alert clinicians to the existence of asymptomatic antibiotic resistance, which is latently present in ICU patients and could be the reason for novel outbreaks of hospital-acquired infections in the future.

## 2. Results

### 2.1. Patient’s Information

All of the patients (n = 60) in the Moscow neuro-ICU were involved in seven-point prevalence surveys in the period from Oct. to Nov. 2019, and each study included 16–24 patients because some patients were admitted to the ICU and then discharged. There were no inclusion or exclusion criteria (Table 1).

Before the study, the patients had been in the neuro-ICU for varying lengths of stay: 1–10 days (n = 45), 11–30 days (n = 6), 31–81 days (n = 7), and 1131 days (n = 1). Therefore, 27 patients were involved in a single survey, 12 patients in 2, 10 patients in 3, 3 patients in 4, 2 patients in 5, 1 patient in 6, and 5 patients in 7 surveys. The median age of the patients was 53.8 years (ranging from 14 to 73 years old), and the sex ratio (male/female) of the patients was 1.07 (31/29). Most patients (n = 28) received carbapenem therapy before/during the surveys, few patients (n = 15) received cephalosporins with/no β-lactamase inhibitors followed by the patients, and the remaining patients (n = 17) were not treated with beta-lactams. Half of the patients (n = 30) had symptoms of gastrointestinal dysfunction (n = 13) and infection of the respiratory system (n = 21), and both symptoms were identified in four patients. Six patients died during this period; the mortality rate was estimated to be 10%. A total of 34 patients were discharged from the neuro-ICU before the end of the study (Figure 1).

### 2.2. Antimicrobial Resistance Genes Detected in Clinical Samples

DNA was extracted from 293 clinical samples, including 146 rectal swabs (r.s.) and 147 tracheal swabs (t.s.). Among the 119 positive antimicrobial resistance (AMR) gene specimens, a total of 357 beta-lactamase genes were detected in DNA samples, including 116 *bla*_TEM_, 79 *bla*_CTX–M_, 45 *bla*_SHV_, 37 *bla*_OXA–48_, 32 *bla*_KPC_, 39 *bla*_NDM_, and 9 *bla*_VIM_. Additionally, 62 integrase genes were identified, including 41 genes belonging to class 1 and 21 genes belonging to class 2 integrase. Most patients carried beta-lactamase genes, *bla*_TEM_ (n = 47), *bla*_SHV_ (n = 19), and *bla*_CTX-M_ (n = 20), and carbapenemase genes, *bla*_NDM_ (n = 14), *bla*_OXA-48_ (n = 13), *bla*_VIM_ (n = 7), and *bla*_KPC_ (n = 10). Additionally, class 1 integrons were detected in 17 patients, and class 2 integrons were detected in 8 patients. Notably, ten patients (namely, 4, 5, 6, 8, 12, 15, 20, 31, 36, and 39) simultaneously carried carbapenemase genes in the trachea and rectum specimens (Figure 1).

Fourteen patients carried carbapenemase genes throughout the study period. Three patients did not have carbapenemase genes at the beginning of the study, but later, they became positive for *bla*_OXA-48_*+bla*_NDM_+*bla*_KPC_, *bla*_OXA-48_+*bla*_VIM_, and *bla*_VIM_*+bla*_NDM_ genes, respectively. In contrast, three patients lost carbapenemase genes during their time in the ICU, and they were not detected at the end of the study. Interestingly, eleven clinical samples of seven patients simultaneously carried three carbapenemase genes, *bla*_NDM_, *bla*_KPC_, and *bla*_OXA-48_, the cephalosporinase gene, *bla*_CTX-M_, and class 1 integrons. A combination of CRGs were also detected: *bla*_NDM_+*bla*_KPC_ was detected in two patients; *bla*_KPC_+*bla*_OXA-48_ in two patients; *bla*_VIM_+*bla*_OXA-48_ in one patient; and *bla*_VIM_+*bla*_NDM_ in one patient. The CRG combination *bla*_OXA-48_+*bla*_NDM_+*bla*_KPC_ was detected in seven patients. The *bla*_OXA-48_+*bla*_NDM_+*bla*_KPC_ CRGs were detected in six patients. Thus, 33 (11%) of the 293 samples were positive for two or more CRGs (Table 2).

In total, 65 (22%) of the 293 clinical samples were positive for CRGs. The *bla*_OXA-48_ gene was detected in 18 (53%) of the 34 CRG-positive r.s. and in 19 (61%) of 31 positive t.s.; the *bla*_NDM_ gene was detected in 22 (64%) rectal and 17 (55%) tracheal samples; the *bla*_KPC_ gene was detected in 17 (50%) rectal and 15 (48%) tracheal samples; and the *bla*_VIM_ gene was detected in 4 (12%) rectal and 6 (19%) tracheal samples. It should be noted that the number of CRGs in the r.s. was two times higher compared to the number in the t.s. The asymptomatic rectal carriage of CRGs was estimated to be present in 17% of the patients, and the asymptomatic tracheal carriage of CRGs was estimated to be present in 18% of the patients (Figure 2).r.

### 2.3. Bacterial Isolates and Asymptomatic Carriage

A total of 344 Gram-negative bacterial isolates were collected from the patients. Overall, 183 isolates were collected from the rectal swabs (r.s.), and 161 isolates were collected from the tracheal swabs (t.s.). *Klebsiella pneumoniae* was the most prevalent bacterium (n = 75 r.s.; n = 68 t.s.), followed by *Escherichia coli* (n = 38 r.s.; n = 23 t.s.), *Acinetobacter baumannii* (n = 21 r.s.; n = 32 t.s.), *Pseudomonas aeruginosa* (n = 24 r.s.; n = 18 t.s.), and *Enterobacter* spp. (n = 8 r.s.; n = 3 t.s.). Less common bacteria included *Stenotrophomonas maltophilia* (n = 1 r.s.; n = 8 t.s.), *Proteus mirabilis* (n = 5 r.s.; n = 2 t.s.), *Citrobacter* spp. (n = 4 r.s.; n = 2 t.s.), *Morganella morganii* (n = 3 r.s.; n = 0 t.s.), *Klebsiella oxytoca* (n = 1 r.s.; n = 1 t.s.), *Klebsiella aerogenes* (n = 1 r.s.; n = 1 t.s.), *Klebsiella variicola* (n = 1 r.s.; n = 0 t.s.), *Serratia marcescens* (n = 0 r.s.; n = 1 t.s.), *Hafnia alvei* (n = 0 r.s.; n = 1 t.s.), *Providencia stuartii* (n = 1 r.s.; n = 0 t.s.), and *Burkholderia gladioli* (n = 0 r.s.; n = 1 t.s.) (Figure 3).

All of the bacterial isolates were collected from 55 patients, while no GNB organisms were found in samples from 5 patients (namely 14, 16, 27, 40, and 60). Bacterial isolates were only collected from the r.s. in six patients (namely 5, 17, 24, 25, 30, and 56), and they were only collected from the t.s. in three patients (namely 33, 51, and 54). The number of isolates collected from one patient varied significantly: 1–10 isolates were obtained from 46 patients, 11–20 isolates were obtained from 6 patients, and 21–30 isolates were obtained from 3 patients.

The majority of isolates (78%) were collected from the patients without clinical manifestation: 141/183 isolates from the r.s. and 126/161 isolates from the t.s. The isolates were identified as *K. pneumoniae* (43%), *E. coli* (18%), *A. baumannii* (14%), and *P. aeruginosa* (13%) and other species, *B. gladioli, Citrobacter* spp., *Enterobacter* spp., *H. alvei, K. aerogenes, K. oxytoca, K. variicola, M. morganii, P. mirabilis, P. stuartii, S. maltophilia*, and *S. marcescens* (12%). The isolates obtained from the patients with gastrointestinal dysfunction and/or respiratory infection were identified as *K. pneumoniae* (38%), *A. baumannii* (21%), *E. coli* (16%), and *P. aeruginosa* (8%) and other species, *Citrobacter* spp., *Enterobacter* spp., *M. morganii, P. mirabilis*, and *S. maltophilia* (17%).

Interestingly, among the isolates obtained from the patients without clinical manifestation, *A. baumannii* was found twice as often in the tracheal swabs compared with the rectal swabs. Moreover, among isolates obtained from the patients without clinical manifestation, *E. coli* was more prevalent in the rectal swabs compared with tracheal swabs (Figure 4).

### 2.4. Antibacterial Resistance Phenotypes and Genotypes

Most isolates (n = 311, 95%) were resistant to beta-lactams, 169 isolates (52%) were resistant to aminoglycosides, 155 isolates (47%) were resistant to fluoroquinolones, 154 isolates (47%) were resistant to chloramphenicol, 26 isolates (34%) were resistant to tetracyclines, and 5 isolates (2%) were resistant to sulfonamides. According to Magiorakos et al. [32], 7 isolates (2%) were attributed to the S category (susceptible), 120 isolates (36%) were attributed to the R category (resistant to at least one agent in <3 antimicrobial functional groups), and 200 isolates (62%) were attributed to the MDR category (resistant to at least one agent in ≥3 antimicrobial functional groups) (Figure 5).

A high rate of MDR isolates was associated with the number of the resistance genes in bacterial isolates. A total of 718 beta-lactamase genes were detected in the isolates, including *bla*_CTX-M_ (n = 212), *bla*_TEM-1_ (n = 156), *bla*_SHV_ (n = 140), *bla*_NDM_ (n = 43), *bla*_OXA-48_ (n = 37), *bla*_KPC_ (n = 25), *bla*_OXA-58-like_ (n = 53), *bla*_OXA-23-like_ (n = 41), *bla*_OXA-40-like_ (n = 9), and *bla*_VIM-2_ (n = 2). Moreover, 262 integrons were detected, including class 1 (n = 248) and class 2 integrons (n = 14).

The isolates susceptible to all antimicrobials (S-category) were identified as *S. maltophilia* (100%); the isolates resistant to at least one agent in <3 antimicrobial functional groups (R-category) were identified as *E. coli* (50%), *A. baumannii* (17%), *Enterobacter* spp. (12%), *Citrobacter* spp. (7%), *K. pneumoniae* (5%), *M. morgannii* (2%), *S. maltophilia* (2%), *B. gladioli* (1%), *H. alvei* (1%), *K. oxytoca* (1%), *P. mirabilis* (1%), and *S. marcescens* (1%); isolates of the MDR phenotype were attributed to *K. pneumoniae* (54%), *P. aeruginosa* (17%), *E. coli* (6%), *P. mirabilis* (2%), *A. baumannii* (19%), *K. aerogenes* (1%), *E. cloacae* (0.5%), and *K. variicola* (0.5%).

It should be noted that 87% (n = 299) of the total bacterial isolates were members of the ESKAPE group, which encompasses the most important antimicrobial-resistant bacterial pathogens. A total of 161 carbapenem-resistant isolates were identified from 38 patients. *K. pneumoniae* was the most common bacterium (40%), followed by *A. baumannii* (34%) and *P.*
*aeruginosa* (22%). Carbapenem-resistant (CR) *A. baumannii* was the most frequent (100% isolates), followed by *P.*
*aeruginosa* (78% isolates)*, K. pneumoniae* (41% isolates), and *E. coli* (2% isolates). Among the carbapenemase genes, *bla*_NDM_ was the most common gene (40%), followed by the *bla*_OXA-48_ (32%), *bla*_KPC_ (26%), and *bla*_VIM-2_ (2%) genes.

### 2.5. Resistomes of K. pneumoniae Clinical Isolates

*K. pneumoniae* isolates were obtained from 39 (65%) patients, including 28 patients who simultaneously carried *K. pneumoniae* in both r.s. and t.s. MDR phenotypes were identified in 98% of *K. pneumoniae* isolates. Eight patients (namely 6,11, 31, 35, 37, 57, 58, and 59) simultaneously carried isolates of *K. pneumoniae* in r.s. and t.s. throughout the observation period, nine patients (namely 2, 9, 10, 13, 15, 29, 32, 34, and 56) carried isolates of *K. pneumoniae* in r.s., and three patients (namely 1, 10, and 55) carried isolates of *K. pneumoniae* in t.s. A total of 145 beta-lactamase genes were detected in the *K. pneumoniae* genomes, including *bla*_SHV_ (n = 143), *bla*_CTX-M_ (n = 142), *bla*_TEM_ (n = 97), *bla*_NDM_ (n = 44), *bla*_OXA-48_ (n = 37), and *bla*_KPC_ (n = 26); class 1 (n = 93) and class 2 (n = 7) integrons. The prevalence of *K. pneumoniae* resistomes combining several genes was detected: *bla*_CTX-M+_*bla*_OXA-48_ (n = 10); *bla*_CTX-M_+*bla*_NDM_ (n = 18); *bla*_CTX-M_+*bla*_OXA-48_+*bla*_KPC_ (n = 5); and *bla*_CTX-M_+*bla*_OXA-48_+*bla*_KPC_+*bla*_NDM_ (n = 18). The gene combination of *bla*_CTX-M_+*bla*_OXA-48_ was detected in *K. pneumoniae* isolates collected from four patients; *bla*_CTX-M_+*bla*_NDM_ was detected in *K. pneumoniae* isolates collected from seven patients; *bla*_CTX-M_+*bla*_KPC_+*bla*_OXA-48_ was detected in *K. pneumoniae* isolates collected from two patients. A novel gene combination that was first identified in this study, *bla*_CTX-M_+*bla*_OXA-48_+*bla*_KPC_+*bla*_NDM_, was detected in *K. pneumoniae* isolates collected from six patients (namely, 3, 4, 7, 8, 11, and 12) (Figure 6).

The latter resistomes were verified using whole-genome sequencing. Genetic determinants coding resistance to aminoglycoside, fosfomycin, chloramphenicol, quinolone, sulfonamide, trimethoprim, and macrolide were identified (Appendix A).

### 2.6. Resistomes of E. coli Clinical Isolates

*E. coli* isolates were obtained from 36 (60%) patients, including 9 patients (namely 23, 39, 42, 43, 44, 46, 48, 49, and 52) who simultaneously carried *E. coli* in both r.s. and t.s. MDR phenotypes were identified for 26% of the *E. coli* isolates. A total of 82 beta-lactamase genes were detected in the *E. coli* genomes including *bla*_CTX-M_ (n = 48), *bla*_TEM-1_ (n = 33), and *bla*_NDM_ (n = 1); and class 1 integrons (n = 21). The prevalence of *E. coli* resistomes combining several genes was detected: *bla*_CTX-M+_*int1* (n = 19) and *bla*_NDM_+*int*1 (n = 1). The gene combination of *bla*_CTX-M+_*int*1 was detected in *E. coli* isolates collected from 13 patients (namely 5, 8, 9, 19, 20, 21, 22, 23, 37, 39, 45, 48, and 49); and *bla*_NDM_+*int1* was detected in *E. coli* isolates collected from 1 patient (namely 23) (Figure 7).

### 2.7. Resistomes of A. baumannii Clinical Isolates

*A*. *baumannii* isolates were obtained from 20 (33%) patients, including 8 patients (namely 1, 4, 8, 10, 15, 22, 34, and 49) who simultaneously carried *A*. *baumannii* in both r.s. and t.s. MDR phenotypes were identified in 85% of the *A*. *baumannii* isolates. In total, 135 beta-lactamase genes were detected in the *A*. *baumannii* genomes, including *bla*_TEM_ (n = 21), *bla*_CTX-M_ (n = 9), *bla*_OXA-51-like_ (n = 54), *bla*_OXA-40-like_ (n = 43), and *bla*_OXA-23-like_ (n = 8); and class 1 integrons (n = 36). The following *A. baumannii* resistomes combining several genes were the most prevalent: *bla*_OXA-40-like_+*bla*_OXA-51-like_+*int1* (n = 23) was collected from 11 patients (namely, 2, 8, 15, 19, 20, 21, 22, 34, 37, 47 and 55). The gene combination of *bla*_OXA-40-like_+*bla*_OXA-51-like_ was detected in 10 *A*. *baumannii* isolates collected from four patients (namely 4, 19, 20, and 34); the gene combination of *bla*_OXA-23-like_+*bla*_OXA-51-like_ was detected in 7 isolates from four patients (namely 1, 12, 32, and 49); the gene combination of *bla*_CTX-M_+*bla*_OXA-40-like_+*bla*_OXA-51-like_+*int1* was detected in 6 isolates from 5 patients (namely 18, 19, 20, 21, and 22). The following gene combinations were rare: *bla*_OXA-51-like_+*int1* was detected in three isolates collected from two patients (namely 9 and 10), *bla*_OXA-40-like_+*bla*_OXA-51-like_ was detected in two isolates from two patients (namely 1 and 46), and *bla*_CTX-M_+*bla*_OXA-23-like_+*bla*_OXA-51-like_ was detected in one isolate from one patient (namely 1) (Figure 8).

### 2.8. Resistomes of P. aeruginosa Clinical Isolates

*P. aeruginosa* isolates were obtained from 14 (23%) patients, including four patients (namely 1, 2, 3, and 32) who simultaneously carried *P. aeruginosa* in both r.s. and t.s. MDR phenotypes were identified for 100% of *P. aeruginosa* isolates. A total of five beta-lactamase genes, including three *bla*_CTX–M_ genes and two *bla*_VIM_ carbapenemase genes, and 32 class 1 integrons were detected in the *P. aeruginosa* genomes. The gene combination of *bla*_CTX-M+_*int*1 was detected in three isolates collected from three patients (namely 2, 3, and 20); *bla*_VIM_+*int*1 was detected in two isolates collected from one patient (namely 34) (Figure 9).

## 3. Discussion

In this study, we estimated the carriage of antimicrobial resistance genes in clinical samples (rectal and tracheal swabs) from 60 patients hospitalized in the neuro-ICU in October-November 2019 with seven-point prevalence surveys. The following beta-lactamase genes were identified in clinical samples: *bla*_TEM_ (47/60), *bla*_SHV_ (19/60), and *bla*_CTX-M_ (20/60); carbapenemase genes (CRGs): *bla*_NDM_ (14/60), *bla*_OXA-48_ (13/60), *bla*_VIM_ (7/60), and *bla*_KPC_ (10/60); and class 1 and 2 integrons: *int1* (17/60) and *int2* (8/60). The following combinations of two CRGs, which occurred simultaneously, were detected: *bla*_NDM_+*bla*_KPC_, *bla*_KPC_+*bla*_OXA-48_, *bla*_VIM_+*bla*_NDM_, and *bla*_VIM_+*bla*_OXA-48_. Moreover, the combination of three CRGs, *bla*_OXA-48_+*bla*_NDM_+*bla*_KPC_, was detected in seven patients. These data are consistent with the findings of the report of Alqahtani et al.; 2021 [33].

Clinical isolates of Gram-negative ESKAPE pathogens (n = 299) were collected from the rectal and tracheal samples, including *K. pneumoniae* (n = 143), *E. coli* (n = 61), *A. baumannii* (n = 53), and *P. aeruginosa* (n = 42). Multi-drug resistant (MDR) phenotypes were identified in 100% of *P. aeruginosa*, 98% of *K. pneumoniae*, 85% of *A. baumannii*, and 26% of *E. coli* samples. Carbapenem-resistant (CR) *A. baumannii* was the most common (100% isolates), followed by *P. aeruginosa* (78% isolates), *K. pneumoniae* (41% isolates), and *E. coli* (2% isolates). It should be noted that most clinical isolates were collected from patients who did not have clinical symptoms; the asymptotical bacterial carriage rate was estimated as 78%. This rate is not too different from the carriage rate reported in an article from Israel in 2013 [34]. Similarly, CR *K. pneumoniae* was the most frequently detected bacteria in Spain and China as an asymptomatic carrier among patients in ICUs [35,36]. In contrast, *E. coli* was the most prevalent agent followed by *K. pneumoniae* in ICUs in Tunisia, Spain, and India [37,38,39]. The most prevalent *A. baumannii* CRGs in our research were *bla*_OXA-51-like_ (100%) and *bla*_OXA-40-like_ (79%), which were different from the most prevalent CRGs in Egypt and Pakistan: *bla*_OXA-23-like_ (78–87%) and *bla*_NDM_ (22–66%) [40,41,42]. In our study, the carriage rate of CRGs in *P. aeruginosa* (two *bla*_VIM_-positive isolates in one patient) was lower compared to the index reported in studies from Uganda, Saudi Arabia, and Egypt (29–50% *bla*_VIM_-positive isolates) [40,43,44].

Among the carbapenemase genes, *bla*_OXA-like_ genes were most commonly identified in the study (77%), including *bla*_OXA-48_ (42%), *bla*_OXA-51-like_ (18%), *bla*_OXA-40-like_ (14%), and *bla*_OXA-23-like_ (3%), followed by *bla*_NDM_ (14%), *bla*_KPC_ (8%), and *bla*_VIM-2_ (1%) genes. The most frequent CRGs in our study were *bla*_NDM_ and *bla*_OXA-like_, which is consistent with reports from China and India [9,39]. *K. pneumoniae*, the most prevalent pathogen in this study, carried the CRGs *bla*_NDM_ (31% isolates), *bla*_OXA-48_ (26% isolates), and *bla*_KPC_ (18% isolates), which differed from the data reported in a study from China: *bla*_NDM_ (25% isolates), *bla*_OXA-48_ (not detected), and *bla*_KPC_ (79% isolates) [45]. A significant rate of *A. baumannii*-associated CRGs was found in our study: *bla*_OXA-51-like_ (100% isolates), *bla*_OXA-40-like_ (81% isolates), and *bla*_OXA-23-like_ (15% isolates). Surprisingly, these rates were different from the data from Poland: 67% of *A. baumannii* isolates carried *bla*_OXA-40-like_ genes, and 33% of isolates carried *bla*_OXA-23-like_ genes [46]. CRGs were rare in *E. coli* isolates: only one *bla*_NDM_ gene was identified in one (2%) isolate, in contrast to the results of a Chinese study: 79% of isolates carried the *bla*_NDM_ gene, and 2% of isolates carried the *bla*_KPC_ gene [45]. The carbapenemase gene, *bla*_VIM_, was detected in 5% of *P. aeruginosa* isolates in our study, in contrast to higher rate of 25% in a study from Pakistan [47].

In our study, the rate of class 1 and 2 integrons’ carriage in MDR-GNB isolated from patients in the neuro-ICU was 76%. These data were somewhat different from reports from France and Uganda (53 and 81%, respectively) [16,48]. Class 1 integrons were more prevalent (n = 248) than class 2 integrons (n = 14) in the isolates in this study. Similar proportions were described in previously published studies from Spain, Iran, and Tunisia [48,49,50]. In this study, the rate of class 1 integrons among CR isolates was 73%, while in reports from Iran, this index was higher (86–95%) [51,52]. In our study, class 1 integrons were detected in 76% of MDR *P. aeruginosa*, 70% of MDR *K. pneumoniae*, 68% of MDR *A. baumannii*, and 34% MDR of *E. coli*. These data are consistent with the findings of reports from Iran and Australia [52,53,54].

In total, the asymptomatic rectal and tracheal carriage of CRGs was found in 17 and 18% of patients, respectively; the rectal and tracheal carriage of class 1 and 2 integrons was found in 23 and 15% of patients, respectively. Such patients may be considered as a potential source of AMR gene transmission in the ICU. Moreover, according to some reports, colonization with MDR potential pathogens and CRGs’ carriage may be a prerequisite for the development of nosocomial infections [34].

Thus, this study highlighted the asymptomatic carriage of carbapenemase genes and the prevalence of potential nosocomial pathogens in the intestine and in the trachea of patients in the neuro-ICU. This is important for clinicians, because it will help them to improve the strategies used for hospital infection control and choose optimal antimicrobial therapies. The novelty of this study is the description of CR-GNB strains that simultaneously carry three carbapenemase genes, *bla*_OXA-48_+*bla*_NDM_+*bla*_KPC_. The limitation of our study was that its single-center nature meant it was impossible to generalize the results. Although the study provided information about the patients regarding prior antibiotic therapy and their length of hospitalization before the study, it did not reveal the importance of these aspects in explaining the reasons for the persistence of resistance genes.

## 4. Materials and Methods

### 4.1. Bioethical Requirements

In this study, we anonymized the data of patients in the ICU. According to the Requirements of the Russian Federation Bioethical Committee, each patient signed informed consent to treatment and laboratory examination. The study was a retrospective observational study. The Burdenko National Medical Research Center of Neurosurgery Review Board approved the study and granted a consent waiver status. Approval Code: #11/2018. Approval Date: 1 November 2018.

### 4.2. Study Design

Seven point-prevalence surveys were conducted weekly among all of the neuro-ICU patients in the period from 18 October 2019 to 29 November 2019. Two clinical samples (rectal and tracheal swabs) were collected from each patient on each survey date. The presence of antimicrobial resistance genes was detected in every clinical sample using real-time PCR (RT-PCR), Gram-negative bacteria were isolated from the samples (Figure 10).

### 4.3. DNA Extraction and AMR Gene Detection in Clinical Samples

Total DNA from clinical samples was extracted using an AmpliPrime DNA-sorb-B reagent kit (InterLabService, Moscow, Russia) in accordance with the manual from the manufacturer. Real-time PCR with specific primers was performed with the obtained DNA preparations to detect beta-lactamase genes—*bla*_TEM_, *bla*_SHV_, *bla*_CTX-M_, *bla*_OXA-48_, *bla*_NDM_, *bla*_VIM_, and *bla*_KPC_—and class 1 and 2 integrons, as described previously [55].

### 4.4. Isolation and Identification of Gram-Negative Bacteria

Gram-negative bacteria were collected from clinical samples through growth in Luria–Bertani (LB) broth (Difco, Sparks, MD, USA) at 37 °C for 18 h and the isolation of single bacterial colonies on Lactose TTC agar with Tergitol-7 (SRCAMB, Obolensk, Russia) and 50 mg/L ampicillin (Thermo Fisher Scientific, Waltham, MA, USA). Bacterial identification was conducted using a MALDI-TOF Biotyper instrument (Bruker, Karlsruhe, Germany). The following ATCC reference strains were used as controls: *Escherichia coli* ATCC 25922, *Pseudomonas aeruginosa* ATCC 27853, and *Klebsiella pneumoniae* ATCC 700603. Bacterial isolates were stored in 15% glycerol at −80 °C.

### 4.5. Susceptibility to Antimicrobials

The minimal inhibitory concentrations (MICs) of antimicrobials belonging to six functional groups: beta-lactams (ampicillin, amoxicillin/clavulanic acid, cefotaxime, ceftazidime, and meropenem), tetracyclines (tigecycline), fluoroquinolones (ciprofloxacin), phenicols (chloramphenicol), aminoglycosides (gentamicin), and sulfonamides (trimethoprim-sulfamethoxazole) were determined via the dilution in agar method using Mueller–Hinton agar (Merck, Darmstadt, Germany) according to the Clinical and Laboratory Standards Institute (CLSI) guidelines, performance standards for antimicrobial susceptibility testing [56]. The results were interpreted according to the European Committee on Antimicrobial Susceptibility Testing. Break-point tables were used for the interpretation of MICs and zone diameters, Version 12.0, 2022 (http://www.eucast.org, accessed on 8 June 2022). *Escherichia coli* strains ATCC 25922 and ATCC 35218 were used for quality control. The criterion for defining multi-drug resistant (MDR) isolates was non-susceptibility to ≥1 agent in ≥3 antimicrobial categories; extensively drug-resistant (XDR) isolates were non-susceptible to ≥1 agent in all but ≤2 categories [32].

The prevalence of bacterial species and antimicrobial resistance phenotypes was calculated using Microsoft Excel v. 1909.

### 4.6. DNA Extraction and PCR Detection of the Resistance Genes in Clinical Isolates

Bacterial thermolysates were used as DNA templates for amplification [57]. Beta-lactamase genes, *bla*_SHV_, *bla*_CTX-M_, *bla*_TEM_, *bla*_OXA-48_, *bla*_KPC_, *bla*_VIM_, and *bla*_NDM_, and class 1 and 2 integrons were detected via PCR using previously described specific primers [57,58,59,60,61,62,63,64].

### 4.7. Whole-Genome Sequencing

The whole-genome sequencing of the isolates was performed using a Nextera DNA Library Preparation Kit and MiSeq Reagent Kit v3 (300 cycles) on an Illumina MiSeq platform. Reads without quality filtering were de novo assembled using Unicycler v 0.4.7 [65] with default parameters. The annotation was performed using the NCBI Prokaryotic Genome Annotation Pipeline [66]. Multilocus sequence typing (MLST) and the identification of antibiotic resistance genes, virulence genes, plasmids, and restriction–modification systems were conducted using the web resource of the Center for Genomic Epidemiology (http://www.genomicepidemiology.org/, accessed on 22 April 2022) and the BIGSDB database (https://bigsdb.pasteur.fr/klebsiella/klebsiella.html, accessed on 22 April 2022).

## 5. Conclusions

The current study focused on the carriage of MDR and ESKAPE Gram-negative bacteria and antimicrobial resistance genes in patients in the neurosurgery-ICU in Moscow in October–November 2019. In total, 55 out of 60 patients harbored significant antimicrobial resistance mechanisms in the neuro-ICU, which implies that infection control measures (contact precautions, etc.) are of critical importance to avoid the spread of resistance. The inappropriate use of antimicrobials should be reduced through antimicrobial stewardship interventions to reduce the effects of antimicrobial resistance gene selection in patients. It was shown that the asymptomatic rectal and tracheal carriage of carbapenem resistance genes was estimated to occur in 17 and 18% of patients, respectively. These patients were a potential source of the transmission of these genes in the ICU. The obtained data indicate the importance of monitoring the asymptomatic carriage of antimicrobial resistance genes, especially carbapenem resistance genes and integrons, and preventing the transmission of such genes within ICUs.

## Figures and Tables

**Figure 1 antibiotics-11-00886-f001:**
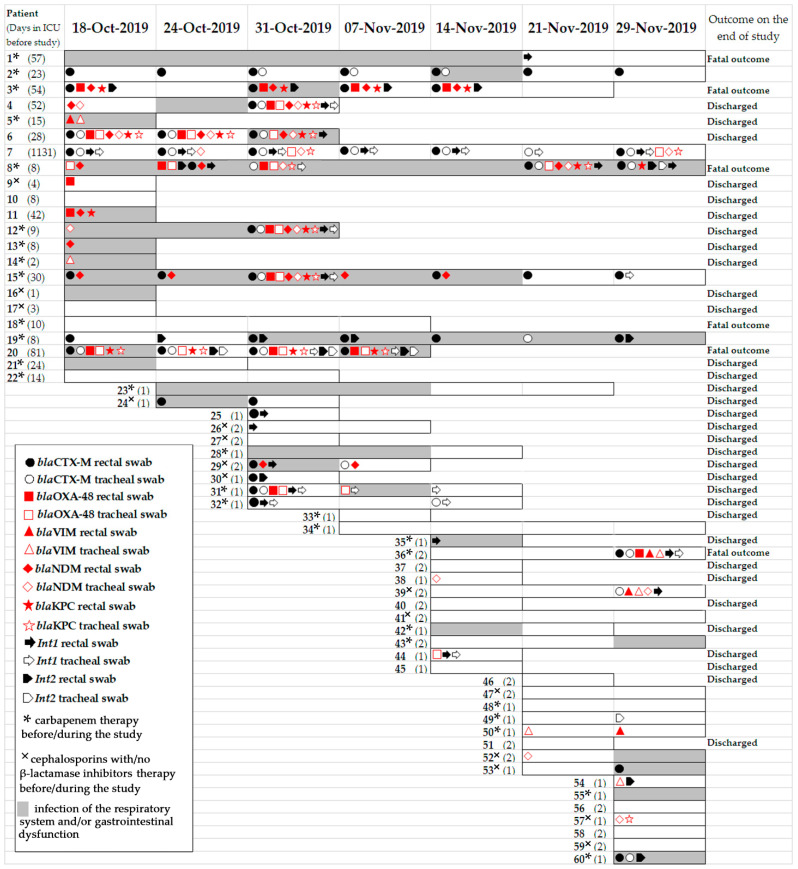
Timeline of beta-lactamase and integron gene detection in rectal and tracheal clinical samples collected from patients in neuro-ICU in October–November of 2019.

**Figure 2 antibiotics-11-00886-f002:**
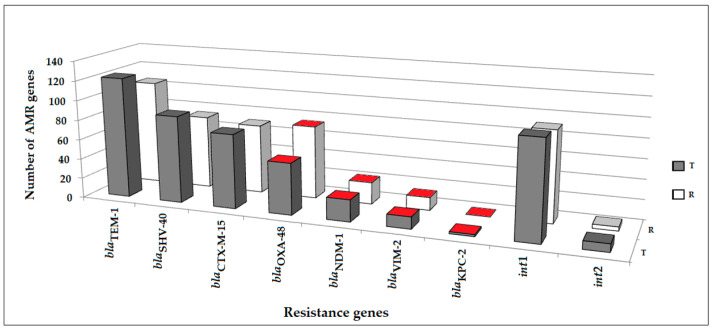
The prevalence of antimicrobial resistance (AMR) genes (n = 419) detected in clinical samples collected from the patients in neuro-ICU: R—rectal swabs, T—tracheal swabs, red color—carbapenem resistance genes.

**Figure 3 antibiotics-11-00886-f003:**
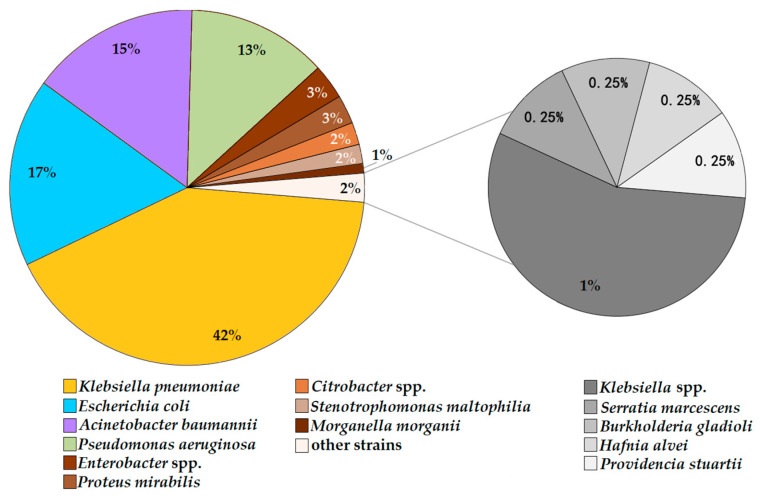
Rate of Gram-negative bacterial isolates on species (n = 344) in the strains in the study collection.

**Figure 4 antibiotics-11-00886-f004:**
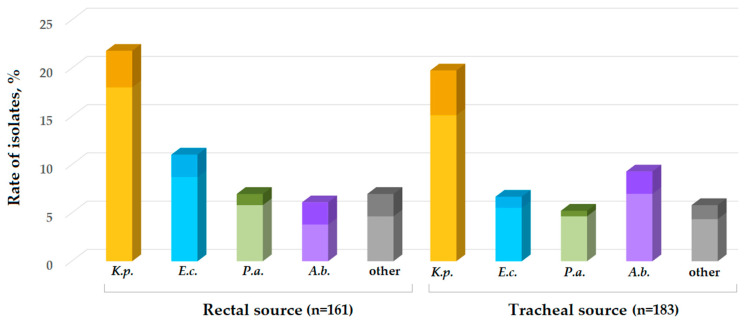
Rate of isolates collected from the patients (rectal and tracheal source) without clinical manifestation (light color) and with infection of the respiratory system or gastrointestinal dysfunction (dark color); *K.p.*—*Klebsiella pneumoniae*, *E.c.*—*Escherichia coli*, *P.a.*—*Pseudomonas aeruginosa*, *A.b.*—*Acinetobacter baumannii*; other—*B. gladioli, Citrobacter* spp., *Enterobacter* spp., *K. aerogenes, K. oxytoca, K. variicola, H. alvei*, *P. mirabilis*, *P. stuartii*, *S. maltophilia*, and *S. marcescens*.

**Figure 5 antibiotics-11-00886-f005:**
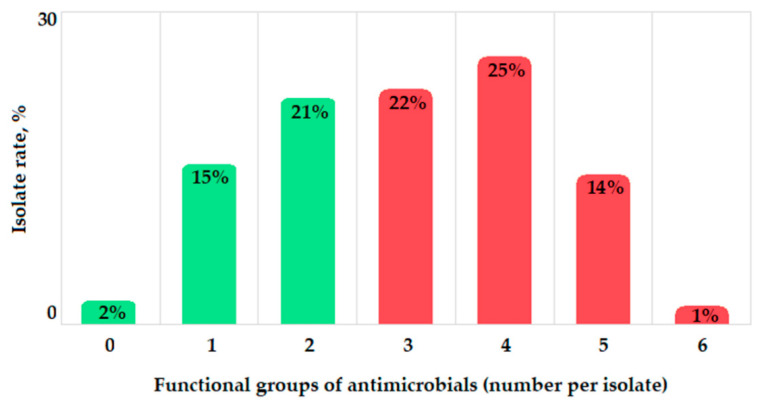
Rate of bacterial isolates resistant to 1–6 antibacterial functional groups: beta-lactams (penicillins, cephalosporins, and carbapenems), tetracyclines, quinolones, phenicols, aminoglycosides, and sulfonamides; S category, green color; MDR category, red color.

**Figure 6 antibiotics-11-00886-f006:**
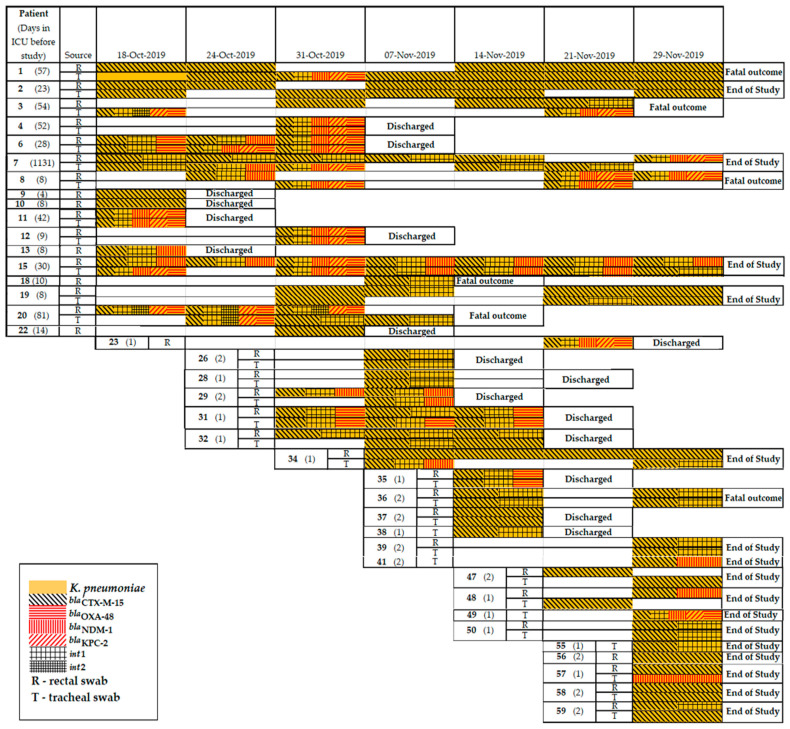
Timeline of *K. pneumoniae* isolation from patients in neuro-ICU in 2019.

**Figure 7 antibiotics-11-00886-f007:**
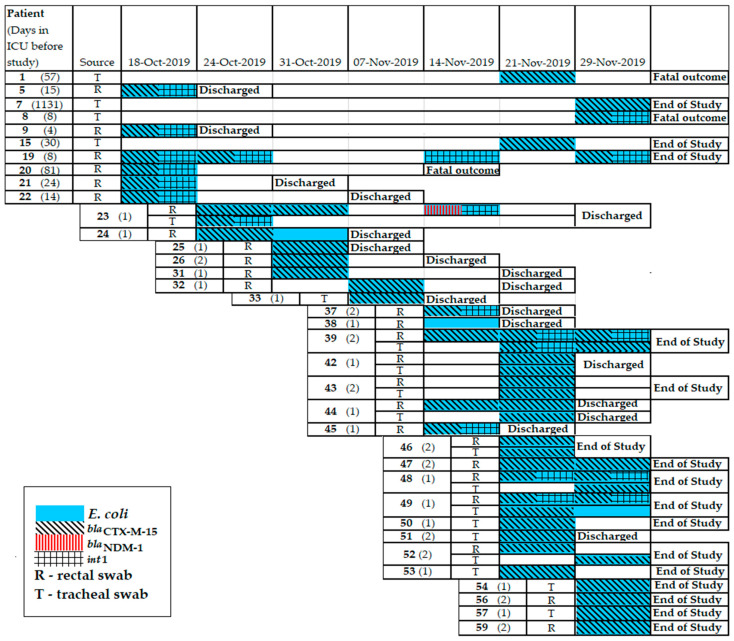
Timeline of *E. coli* isolation from patients in neuro-ICU in 2019.

**Figure 8 antibiotics-11-00886-f008:**
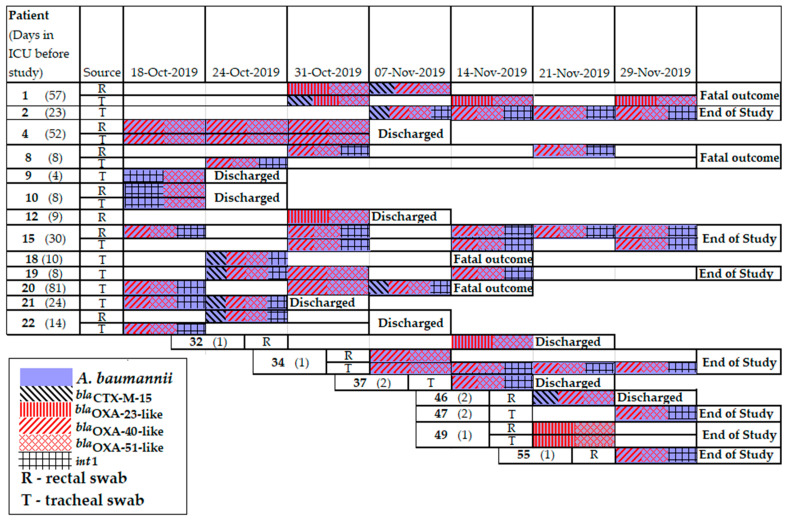
Timeline of *A. baumannii* isolation from patients in neuro-ICU in 2019.

**Figure 9 antibiotics-11-00886-f009:**
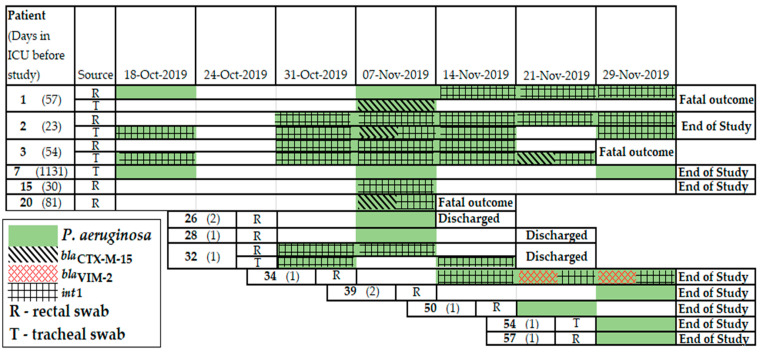
Timeline of *P. aeruginosa* isolation from patients in neuro-ICU in 2019.

**Figure 10 antibiotics-11-00886-f010:**
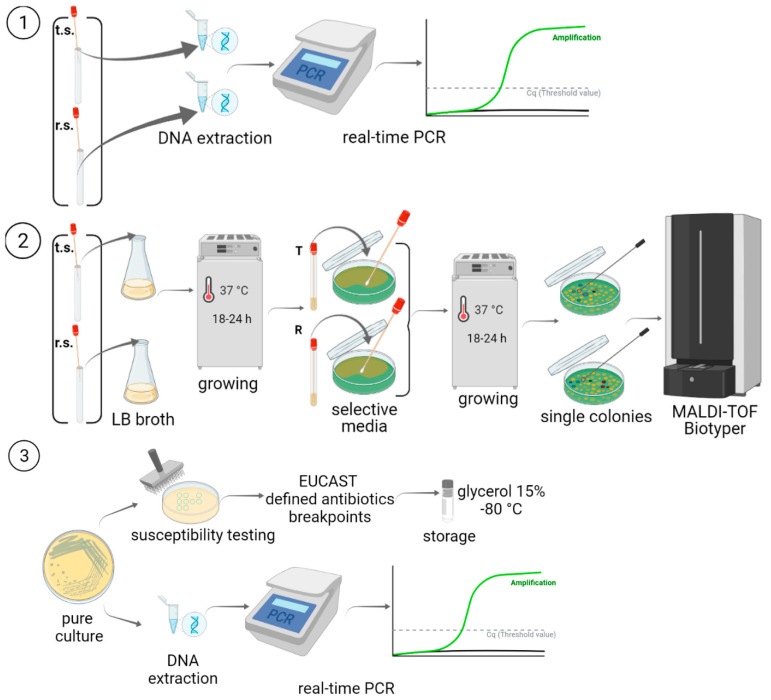
Study design: **①**—DNA extraction and AMR gene detection in clinical samples; **②**—isolation and identification of Gram-negative bacteria from clinical samples; **③**—antimicrobial susceptibility testing, DNA extraction, PCR detection of the resistance genes, and storage.

**Table 1 antibiotics-11-00886-t001:** Patient’s information.

Item	Dates of Point-Prevalence Surveys/Number of Patients Per Survey						
	18 October 2019	24 October 2019	31 October 2019	07 October 2019	14 October 2019	21 November 2019	29 November 2019
Total patients	22	16	23	17	23	21	24
Respiratory infection	8	6	6	6	5	2	6
Gastrointestinal dysfunction	4	2	4	2	1	0	1
Carbapenem therapy	13	10	13	13	15	14	13
Cephalosporins with/no β-lactamase inhibitors therapy	3	1	5	2	2	5	7
Admission to the ICU	22	2	8	2	11	8	7
Fatal outcome	0	0	0	0	2	1	3
Discharged	0	8	1	8	3	9	1

**Table 2 antibiotics-11-00886-t002:** Trends in the content of resistance genes in patients.

Trends in Carbapenemase Gene Content	Patients
Patients carried carbapenemase genes both in r.s. and t.s. simultaneously	4, 5, 6, 8, 12, 15, 20, 31, 36, 39
Patients positive on carbapenemase genes during the study	3, 5, 6, 8, 9, 11, 13, 14, 20, 38, 44, 50, 54, 57
Patients became positive for carbapenemase genes during the study	7, 36, 39
Patients became negative for carbapenemase genes during the study	15, 31, 52
Patients positive on *bla*_NDM_+*bla*_KPC_+ *bla*_OXA-48_+*bla*_CTX-M_+*int1*	r.s. 3, 4, 6, 8, 11,12, 15 t.s. 4, 6, 7, 8, 12, 15
Patients positive on *bla*_NDM_+*bla*_KPC_	r.s. 6 t.s. 57
Patients positive on *bla*_KPC_+*bla*_OXA-48_	r.s. 8, 20 t.s. 20
Patients positive on *bla*_VIM_+*bla*_OXA-48_	r.s. 36
Patients positive on *bla*_VIM_+*bla*_NDM_	t.s. 36
Patients positive on *bla*_OXA-48_+*bla*_NDM_+*bla*_KPC_	r.s. 4, 6, 8, 12, 11, 15 t.s. 4, 6, 7, 8, 12, 15

Note: r.s.—rectal swab, t.s.—tracheal swab.

## Data Availability

The data presented in this study are openly available in the GenBank database at accession numbers [JAGUTV000000000, JAGUTT000000000, JAGUTR000000000, JAGUTQ000000000, JAGVVJ000000000, JAHAVL000000000].

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
