# Peer review of "Rectal and Tracheal Carriage of Carbapenemase Genes and Class 1 and 2 Integrons in Patients in Neurosurgery Intensive Care Unit"

_antibiotics, 2022, doi:10.3390/antibiotics11070886_

Round 1

Reviewer 1 Report

This is an interesting study from a neuro-ICU in Russia where tracheal and stool cultures from 60 patients in a point-prevalence study were obtained. The study provides data on microbiology and, importantly, on genetic elements providing antimicrobial resistance. Here are some comments that could improve the manuscript:

1.       Line 50: ‘asymptotical’: maybe you mean asymptomatic

2.       Line 54: the first two measures are actually infection control measures. Thus, please rephrase so to specifically include the term ‘infection control measures’ (do not delete the specific examples you mentioned)

3.       Introduction: Some data regarding the prevalence of antimicrobial resistance in Russia from older studies or other sources (for example, in Europe, ECDC collects and presents such data online) should be included, even if they are just descriptive and do not provide mechanisms of resistance. For example, doi: 10.1016/j.jgar.2019.09.008

4.       Line 69: Define what you mean by ‘intestinal dysfunction’

5.       Line 79 and elsewhere: Major? What do you mean? Maybe ‘most patients’?

6.       Line 87: ‘accepted’? Probably ‘became positive for’

7.       Line 89: ‘spent’, not ‘spend’

8.       English needs revision

9.       Line 90: Interestingly, not interesting

1-   Line 99 and 100: ‘asymptotic’: maybe you mean asymptomatic

1-   Line 120: bacterium, not bacteria

1-   Line 132: While reading this sentence it is not clear if cultures were obtained while actively looking for microorganisms or if this was waived due to a short stay in the ICU. So, this sentence could be rephrased to something like this ‘…while in 5 patients, cultures did not yield any microorganisms’

1-   Line 171: It is not clear when you read this sentence what S- and R-categories are. Please define

1-   Line 206: ‘phenicol’? Maybe you mean chloramphenicol?

1-   A limitation subsection should be added at the end of the discussion section. For example, this is a single-center study, which limits the generalization of the results. Furthermore, there are no data on previous antimicrobial exposure, time of hospitalization of the patients, or any other data that could relate to the reasons for harboring such resistance genes

1-   Line 310: You could simply say that data was anonymized

1-   Line 312: Since patients were in the neuro-ICU (having neurological damage), I guess that some (or most) patients could not sign an informed consent document. Thus, probably a next of kin would do that. Please state it, if this is the case

1-   Methods: You should add definitions for and MDR, XDR. I think this is a popular reference that defines them doi: 10.1111/j.1469-0691.2011.03570.x

1-   Figure 5 legend: What do you mean with ‘fenicols’?

2-   Figure 5 footnote and figure 2 at the y-axis: what do you mean with ‘pcs’?

2-   One of the important messages of this manuscript is that 55 out of 60 patients harbored significant antimicrobial resistance mechanisms in the neuro-ICU, and this implies that infection control measures (contact precautions etc) are of critical importance to avoid resistance spread. Furthermore, inappropriate use of antimicrobials should be reduced through antimicrobial stewardship interventions to reduce the push of antimicrobial resistance gene selection in patients. I feel that these are important to be noted at the end of the discussion section and maybe at the conclusions as well (some of them are implied after line 370 in the conclusions section)

Author Response

We are grateful to Reviewer 1 for the positive assessment of our manuscript. We can answer the questions as follows:

Point 1: Line 50: ‘asymptotical’: maybe you mean asymptomatic

Response 1: Thank you for your careful reading. We changed ‘asymptotical’ on ‘asymptomatic’ in the line 50.

Point 2: Line 54: the first two measures are actually infection control measures. Thus, please rephrase so to specifically include the term ‘infection control measures’ (do not delete the specific examples you mentioned)

Response 2: According to the Reviewer’s suggestion, we added the term ‘infection control measures’ in the text.

Point 3: Introduction: Some data regarding the prevalence of antimicrobial resistance in Russia from older studies or other sources (for example, in Europe, ECDC collects and presents such data online) should be included, even if they are just descriptive and do not provide mechanisms of resistance. For example, doi: 10.1016/j.jgar.2019.09.008

Response 3: The sentence has been corrected as follows: ‘Over the past 15 years, several single and multicenter MDR-GNB hospital-acquired infections microbiological surveillance studies have been conducted in the Russian Federation [24-29]. These studies contained only descriptive reports without highlighting the mechanisms of antimicrobial resistance and colonization of the patients by the MDR-GNBs and CRGs.’ The references were added in to the References section.

Point 4: Line 69: Define what you mean by ‘intestinal dysfunction’

Response 4: We clarified the symptom and replaced ‘intestinal dysfunction’ on ‘gastrointestinal dysfunction’. Gastrointestinal dysfunction means the return of food through a feeding tube or a disorder of gastrointestinal motility, when drug stimulation of peristalsis is required.

Point 5: Line 79 and elsewhere: Major? What do you mean? Maybe ‘most patients’?

Response 5: Thank you for your careful reading. We changed ‘Major’ on ‘most’ in lines 79, 139, 159, 182, 265 and 270.

Point 6: Line 87: ‘accepted’? Probably ‘became positive for’

Response 6: Thank you for your careful reading. We changed ‘accepted’ on ‘became positive for’ in line 87.

Point 7: Line 89: ‘spent’, not ‘spend’

Response 7: It was done.

Point 8: English needs revision

Response 8: According to the Reviewer’s suggestion, we checked English by web-recourse ‘ProWritingAid’ and native speaker.

Point 9:  Line 90: Interestingly, not interesting

Response 9: It was done.

Point 10: Line 99 and 100: ‘asymptotic’: maybe you mean asymptomatic

Response 10: Thank you for your careful reading. We changed ‘asymptotical’ on ‘asymptomatic’ in lines 99 and 100.

Point 11: Line 120: bacterium, not bacteria

Response 11: It was done.

Point 12: Line 132: While reading this sentence it is not clear if cultures were obtained while actively looking for microorganisms or if this was waived due to a short stay in the ICU. So, this sentence could be rephrased to something like this ‘…while in 5 patients, cultures did not yield any microorganisms’

Response 12: According to the Reviewer’s suggestion, we rephrased the sentence: ‘All bacterial isolates were collected from 55 patients, while no GNB organisms were found in samples from 5 patients (namely, 14, 16, 27, 40, and 60).’

Point 13: Line 171: It is not clear when you read this sentence what S- and R-categories are. Please define

Response 13: According to the Reviewer’s suggestion, we corrected the sentence: ‘The isolates susceptible to all antimicrobials (S-category) were identified as S. maltophilia (100 %); the isolates resistant to at least 1 agent in < 3 antimicrobial functional groups (R-category) – E. coli (50 %), A. baumannii (17 %), etc.’

Point 14: Line 206: ‘phenicol’? Maybe you mean chloramphenicol?

Response 14: Thank you for your careful reading. We changed ‘phenicol’ on ‘chloramphenicol’ in the line 206.

Point 15: A limitation subsection should be added at the end of the discussion section. For example, this is a single-center study, which limits the generalization of the results. Furthermore, there are no data on previous antimicrobial exposure, time of hospitalization of the patients, or any other data that could relate to the reasons for harboring such resistance genes

Response 15: According to the Reviewer’s suggestion, we added the phrase ‘Most patients (n=28) received carbapenem therapy before/during the surveys, few patients (n=15) received cephalosporins with/no β-lactamase inhibitors followed by the patients, and the remaining patients (n=17) were not treated by beta-lactams.’ in the ‘2.1. Patient’s information’ section.

Moreover, we added the phrase ‘The limitations of our single-center study were impossible in the generalization of the results. Although the study provides the information on prior antibiotic therapy and the length of hospitalization before the study, it does not reveal its importance in explaining the reasons for the persistence of resistance genes.’ at the end of the discussion section.

Point 16: Line 310: You could simply say that data was anonymized

Response 16: We changed the phrase ‘Clinical isolates described in this study were marked without the name, date of birth, address, number of the disease history, personal documents, and other personal materials of the patients.’ to the sentence ‘In this study, we anonymized the data of the patients of ICU.’

Point 17: Line 312: Since patients were in the neuro-ICU (having neurological damage), I guess that some (or most) patients could not sign an informed consent document. Thus, probably a next of kin would do that. Please state it, if this is the case

Response 17: On this point, we can explain the followings: All patients sign an informed consent prior to surgery, as long as they are conscious and can be responsible for their actions. They became neurologically damaged after the surgery and then admitted to the ICU. The Institute provides planned neurosurgical care.

Point 18: Methods: You should add definitions for and MDR, XDR. I think this is a popular reference that defines them doi: 10.1111/j.1469-0691.2011.03570.x

Response 18: According to the Reviewer’s suggestion, we added MDR and XDR definition in 4.5 subsection of the ‘Materials and Methods’ section: ‘Criteria for defining multi-drug resistant (MDR) isolates was non-susceptible to ≥1 agent in ≥3 antimicrobial categories, extensively drug resistant (XDR) was non-susceptible to ≥1 agent in all but ≤2 categories’.

Point 19: Figure 5 legend: What do you mean with ‘fenicols’?

Response 19: Thank you for the correction; we changed the ‘fenicols’ to ‘phenicols’.

Point 20: Figure 5 footnote and figure 2 at the y-axis: what do you mean with ‘pcs’?

Response 20: We corrected the Y-axis in Figure 2: ‘Number of AMR genes’ and caption to Figure 2: ‘The prevalence of antimicrobial resistance (AMR) genes (n = 419) detected in clinical samples collected from the patients of neuro-ICU: R – rectal swabs, T – tracheal swabs.’

We corrected the X-axis name in Figure 5 as follows: ‘Functional groups of antimicrobials (number per isolate)’.

Point 21: One of the important messages of this manuscript is that 55 out of 60 patients harbored significant antimicrobial resistance mechanisms in the neuro-ICU, and this implies that infection control measures (contact precautions etc) are of critical importance to avoid resistance spread. Furthermore, inappropriate use of antimicrobials should be reduced through antimicrobial stewardship interventions to reduce the push of antimicrobial resistance gene selection in patients. I feel that these are important to be noted at the end of the discussion section and maybe at the conclusions as well (some of them are implied after line 370 in the conclusions section)

Response 21: According to the Reviewer’s suggestion, we added the phrase ‘The fifty-five out of 60 patients harbored significant antimicrobial resistance mechanisms in the neuro-ICU, which implies that infection control measures (contact precautions etc.) are of critical importance to avoid the spread of resistance. Inappropriate use of antimicrobials should be reduced through antimicrobial stewardship interventions to reduce the push of antimicrobial resistance gene selection in patients.’ into the Conclusion section.

Reviewer 2 Report

Interesting how 9 authors have written a real manuscript of 12 pages in MDPI format (2/3 occupied space from a page).

Instructions for authors must be checked and applied regarding, text size, text settings, etc.

Keywords must reflect the main characteristic words of the paper (usually reflected also by the title) in the best way to increase the paper's relevance and chances to be find when searching it after key words. So, for the actual title, I suggest the following keywords:

rectal and tracheal carriage; carbapenemase genes; class 1 and 2 integrons; neurosurgery; intensive care unit; + other keywords the authors consider most important - up to 10. 

Introduction – text must be justified set.

Introduction is too short. I suggest better developing it, detailing the bacteria that can be found in an intensive care unit (as it is very well developed in https://doi.org/10.1016/j.scitotenv.2019.06.076  ) and in surgery departments (https://doi.org/10.3390/antibiotics.9020081 )

Aim of the study, L59-60, is extremely poor. Please make the aim of the study a separate, last paragraph of Introduction section (to make it easier visible for those interested in the topic), highlighting better some aspects by responding to the following questions: What makes special this study? Which is its novelty character or its special aspects? Why have the author chosen this topic? What differentiate this paper from others already published in the same/similar topic? Already published studies are done in this field, so it is important to emphasize what does your study brings special. 

L63-64. I am sure that the authors understand that the no. of patients is much too small for a relevant statistic. In order to have a statistically significant statistic, each group must have at least 33 individuals/samples (whether it's experimental or clinical trials).

Section 2.1. I suggest tabulating it. Much easier to follow. Or a flow chart, describing criteria for patients’ selection (inclusion/exclusion criteria), as well as their division into groups. The authors will decide how they consider best shape for presenting these data (but simple text is confusing and hard to follow). Moreover, patients’ selection belongs to Material and methods section (not to Results) – please move it in the proper place.

Same tabulate needing for L74-109.

After L307, please provide a strong paragraph describing the strengths and the weakness of this research.

Section 4. Which statistic program and its variant were used for data processing?

Author Response

We are grateful to Reviewer 2 for the positive assessment of our manuscript. We can answer the questions as follows:

Point 1: Interesting how 9 authors have written a real manuscript of 12 pages in MDPI format (2/3 occupied space from a page).

Response 1: On this point, we can explain that different groups were involved in the study: clinical physicians (E.O.N., K.N.V., and S.I.A.), a microbiologist (N.T.S.), molecular microbiologists (K.E.S., A.E.I., and F.G.N.), a genomic analytic (K.A.A.) and a scientific supervisor (F.N.K.). All authors contributed significantly to the research and writing of the manuscript.

Point 2: Instructions for authors must be checked and applied regarding, text size, text settings, etc.

Response 2: We checked Instructions for authors and used a text template.

Point 3: Keywords must reflect the main characteristic words of the paper (usually reflected also by the title) in the best way to increase the paper's relevance and chances to be find when searching it after key words. So, for the actual title, I suggest the following keywords: rectal and tracheal carriage; carbapenemase genes; class 1 and 2 integrons; neurosurgery; intensive care unit; + other keywords the authors consider most important - up to 10.

Response 3: According to the Reviewer’s suggestion, we changed keywords: ‘rectal and tracheal carriage; carbapenemase genes; class 1 and 2 integrons; neurosurgery; intensive care unit; gram-negative bacteria; Klebsiella pneumoniae; Escherichia coli; Acinetobacter baumannii; Pseudomonas aeruginosa’.

Point 4: Introduction – text must be justified set. Introduction is too short. I suggest better developing it, detailing the bacteria that can be found in an intensive care unit (as it is very well developed in https://doi.org/10.1016/j.scitotenv.2019.06.076) and in surgery departments (https://doi.org/10.3390/antibiotics.9020081)

Response 4: According to the Reviewer’s suggestion, we have expanded the Introduction section and added some appropriated references.

Point 5: Aim of the study, L59-60, is extremely poor. Please make the aim of the study a separate, last paragraph of Introduction section (to make it easier visible for those interested in the topic), highlighting better some aspects by responding to the following questions: What makes special this study? Which is its novelty character or its special aspects? Why have the author chosen this topic? What differentiate this paper from others already published in the same/similar topic? Already published studies are done in this field, so it is important to emphasize what does your study brings special.

Response 5: According to the Reviewer’s suggestion we rewrote the aim of the study: ‘In order to provide possible future intervention for clinically rational antibiotic usage, the aim of this study was to evaluate the rate of rectal and tracheal carriage of MDR-GNB and CRGs in the patients of Moscow neuro-ICU. The novelty of this study is the description of “heavily armed” CR-GNB strains that have not been reported before this study. The results of this study are intended to alarm clinicians for the existence of asymptomatic antibiotic resistance, which is latently presents in ICU patients and could be the reason for novel outbreaks of hospital-acquired infections in the future.

Point 6: L63-64. I am sure that the authors understand that the no. of patients is much too small for a relevant statistic. In order to have a statistically significant statistic, each group must have at least 33 individuals/samples (whether it's experimental or clinical trials).

Response 6: All patients (n=60) of Moscow neuro-ICU were involved in 7 point-prevalence surveys in the period from Oct. to Nov. 2019, while each study included 16-24 patients because some patients were admitted into the ICU and discharged from this. There were not any inclusion/exclusion criteria applied. The patients were not separated into any groups because they were not any experimental or clinical trials in the study.

Point 7: Section 2.1. I suggest tabulating it. Much easier to follow. Or a flow chart, describing criteria for patients’ selection (inclusion/exclusion criteria), as well as their division into groups. The authors will decide how they consider best shape for presenting these data (but simple text is confusing and hard to follow). Moreover, patients’ selection belongs to Material and methods section (not to Results) – please move it in the proper place.

Response 7: According to the Reviewer’s suggestion, we added Figure 10, describing study design in the Material and methods section, and Table 1 in the 2.1. Subsection of the Results section.

Point 8:  Same tabulate needing for L74-109.

Response 8: According to the Reviewer’s suggestion, we added table 2.

Point 9: After L307, please provide a strong paragraph describing the strengths and the weakness of this research.

Response 9: We added the text ‘Thus, this study highlighted the fact of asymptomatic carriage of carbapenemase genes and prevalence of potential nosocomial pathogens in the intestine and in the trachea of neuro-ICU patients. This is important for clinicians because it will help them to improve the strategy for hospital infection control and choose optimal antimicrobial therapy. The novelty of this study is a description of CR-GNB strains carrying three carbapenemase genes blaOXA-48+blaNDM+blaKPC simultaneously. The limitations of our single-center study were impossible in the generalization of the results. Although the study provides the information on prior antibiotic therapy and the length of hospitalization before the study, it does not reveal its importance in explaining the reasons for the persistence of resistance genes.’ into Discussion section.

Point 10: Section 4. Which statistic program and its variant were used for data processing?

Response 10: We added information in the 4.5. Subsection of Material and methods section: ‘The prevalence of bacterial species and antimicrobial resistance phenotypes were calculated using Excel Microsoft program v. 1909.’

Round 2

Reviewer 1 Report

The manuscript has been improved during the revision process

Author Response

We are grateful to Reviewer 1 for the positive assessment of our manuscript. According to the Reviewer’s suggestion, the text has been checked for correct use of grammar and common technical terms by MDPI.

Reviewer 2 Report

The quality of the figures is from poor to very poor, some of them are blurred, and some of them unreadable. I think that the authors saved them in a not proper format. Just print screen-ing their original form would be helpful.

Also, use black colour when writing on the figures, not grey.

Please provide best quality/clarity figures.

Author Response

We are grateful to Reviewer 2 for the positive assessment of our manuscript. According to the Reviewer’s suggestion, the text has been checked for correct use of grammar and common technical terms by MDPI. The quality of the figures was improved and the figures were saved in proper format (.tiff).